# Using pose estimation to identify regions and points on natural history specimens

**Yichen He**[1]*, **Christopher R. Cooney**[1], **Steve Maddock**[2], **Gavin H. Thomas**[1,3]

**1** Ecology and Evolutionary Biology, School of Biosciences, University of Sheffield; Alfred Denny Building, University of Sheffield, Sheffield, United Kingdom, **2** Department of Computer Science, University of Sheffield; Regent Court, University of Sheffield, Sheffield, United Kingdom, **3** Bird Group, Department of Life Sciences, The Natural History Museum at Tring; Tring, United Kingdom

* csyichenhe@gmail.com

## Abstract

A key challenge in mobilising growing numbers of digitised biological specimens for scientific research is finding high-throughput methods to extract phenotypic measurements on these datasets. In this paper, we test a pose estimation approach based on Deep Learning capable of accurately placing point labels to identify key locations on specimen images. We then apply the approach to two distinct challenges that each requires identification of key features in a 2D image: (i) identifying body region-specific plumage colouration on avian specimens and (ii) measuring morphometric shape variation in *Littorina* snail shells. For the avian dataset, 95% of images are correctly labelled and colour measurements derived from these predicted points are highly correlated with human-based measurements. For the *Littorina* dataset, more than 95% of landmarks were accurately placed relative to expert-labelled landmarks and predicted landmarks reliably captured shape variation between two distinct shell ecotypes ('crab' vs 'wave'). Overall, our study shows that pose estimation based on Deep Learning can generate high-quality and high-throughput point-based measurements for digitised image-based biodiversity datasets and could mark a step change in the mobilisation of such data. We also provide general guidelines for using pose estimation methods on large-scale biological datasets.

## Author summary

As the digitisation of natural history collections continues apace, a wealth of information is waiting to be mobilised from these vast digital datasets that can help address many evolutionary and ecological questions. Deep Learning has achieved success on many real-world tasks such as face recognition and image classification. Here, we use deep learning to measure phenotypic traits of specimens by placing points on photos of birds and periwinkles. We show that the measurements produced by Deep Learning are generally accurate and very similar to manual measurements taken by experts. As Deep Learning methods vastly reduce the time required to produce these measurements, our results demonstrate the great potential of Deep Learning for future biodiversity studies.

**Data Availability Statement:** Specimen images, expert labelled images and machine-predicted coordinates associated with training and testing our deep learning model are available https://doi.org/10.15131/shef.data.19432469. All analysis

data is available in the manuscript. The code is available on https://github.com/EchanHe/Pose_DL_specimen_pts.

**Funding:** This work was funded by a Leverhulme Early Career Fellowship (ECF-2018-101) and Natural Environment Research Council Independent Research Fellowship (NE/T01105X/1) to Christopher R. Cooney; a European Research Council grant (615709, Project 'ToLERates') and Royal Society University Research Fellowship (UF120016, URF\R\180006) to Gavin H. Thomas; and a Leverhulme Centre for Doctoral Training grant to Yichen He. The funders had no role in study design, data collection and analysis, decision to publish, or preparation of the manuscript.

**Competing interests:** The authors have declared that no competing interests exist.

## Introduction

Measurement of phenotypic traits from large collections of digitised specimens is an increasingly important step in biodiversity studies that address evolutionary and ecological questions. While generating genomic data has become progressively more cost-effective, allowing the compilation of complete genome databases for numerous organisms [1,2], phenotypic databases are less well developed in part due to low efficiency in collecting phenotypic information [3,4]. A significant challenge is to build high-throughput phenotyping pipelines for such large-scale data. Globally, natural history museums house extensive, but often underexploited, collections of biological specimens. Measurement is often limited by access to and quality of specimens, yet there are an estimated 1.2 to 1.9 billion specimens in museum collections globally [5]. Collection digitisation is a major goal for many natural history museums and can provide a rich source of phenotypic and biodiversity data [6–8].

Digitisation (including 2D photos, videos and 3D scans) and collecting associated metadata is a straightforward way to store permanent records of specimens as digital data [8–13]. Indeed, many natural history museums and researchers are actively digitising large proportions of existing natural history collections [7,14–16], and major international projects have made significant advances in developing digitisation infrastructure towards the goal of digitising and sharing entire collections. However, most of the raw digitised data (e.g. images, scans) emerging from such digitisation efforts cannot readily be used in ecological and evolutionary analyses without extensive processing. To mobilise digital natural history data, robust and high-throughput data extraction (e.g. phenotyping or trait measurements) pipelines are necessary.

A particular challenge is the placement of points on digitised images, where point annotation is used for measuring phenotypic traits on digital data in a variety of ways. Specifically, some form of point placement is required for identifying key features on digitised specimens. Perhaps the most common application is in the field of geometric morphometrics where point placement is used to annotate (typically) homologous landmarks for the measurement of shape [17,18]. Applications span neontological and paleontological specimens, with the raw data for such studies commonly derived from natural history collections [14,19–21]. Other applications of point placement include tracking animal behaviours or motions [22–24] and identifying the location of appendages and/or body regions within images of organisms [15]. To date, the majority of studies rely on experts for point annotation. However, manual point placement is time-consuming and labour-intensive, especially as datasets become increasingly large. Seeking ways that can increase labelling speed while maintaining accuracy is therefore a growing challenge for biodiversity science.

Computer vision offers considerable promise for mobilising phenotypic data from natural history collections [4]. The field of computer vision focuses on developing and applying algorithms to make computers detect features on digital images and videos and to use the features to solve vision-related problems such as image classification and image segmentation. The current state-of-the-art for many computer vision tasks is Deep Learning. In the context of computer vision, Deep Learning models have been developed to extract image features automatically from pre-labelled training sets. Deep Learning uses convolutional neural networks (CNN) to overcome the complexity of input images or videos [25]. CNNs take input images and repeatedly apply filters to create feature maps to differentiate among feature types. The major advance of CNNs is the potential to automatically learn large numbers of features in parallel that are specific to the prediction problem, rather than requiring hand-crafted, predefined and task-specific image features (e.g. edges, corners) to locate key points [26,27]. The use of convolutional layers provides learning parameters that can be shared across inputs,

increasing the network performance by reducing overfitting and computational costs (reducing the total number of parameters). After many convolutional layers, different levels of image features are automatically extracted from the training set. The extracted features can then be used in Deep Learning models for a range of different vision tasks. There is growing interest and research into the application of such methods in ecology and evolutionary biology [4,28–32].

Here, for the task of automated point annotation we focus on pose estimation methods based on Deep Learning. Pose estimation uses deep neural networks and was developed to identify and label human body parts and joints as points on images. Numerous variants of pose estimation models have been developed. In particular, the Stacked Hourglass [33], Deep-Cut [34] and Convolutional Pose Machine [35] approaches have been shown to perform well on benchmark human pose estimation datasets [36,37]. Indeed, the use of pose estimation in biological data processing and analysis has been growing rapidly and previous studies have used pose estimation algorithms to detect landmarks to track animals' positions or postures in videos [22,38–41] and quantify specimens' morphological traits [42]. Digitised natural history collections represent a valuable data source for many biological analyses, yet the potential for pose estimation to locate points and extract useful data from such datasets has yet to be fully explored. We therefore explored the performance of pose estimation (i.e. whether points identifying focal body locations/regions can be placed accurately on images) on digitised specimens, particularly in cases where the subjects have variable appearance but similar postures, as is the case in many digital image-based natural history datasets.

We focused our analysis on two datasets that pose subtly different computational challenges. First, we apply and assess the performance of pose estimation methods to locate body regions and extract colour information from standardised digital images of birds. The avian specimen dataset is part of an extensive set of photos of bird specimens representing >85% of all of the world's bird species taken under controlled lighting conditions at the Natural History Museum, Tring [43]. The primary goal of a high-throughput colour measuring pipeline is to apply an automated body region detection algorithm to allow measurement of plumage colouration from specific bird body regions. Here, the accuracy of point placement is of secondary importance to the accuracy of colour measurement. Thus, placement of points only needs to be within the broader body region, rather than at a specific location within the body region. Second, we examine the performance of placing points on hundreds of images of shells of the marine gastropod *Littorina saxatilis* (periwinkle). Here, the goal is to use the points as morphometric landmarks in downstream geometric morphometric analysis. We expect the features of the *Littorina* dataset to be more readily distinguishable than those from the bird plumage data. However, geometric morphometric analysis also likely requires a higher degree of accuracy in point placement. We first focus on the avian specimen dataset and explore a wide range of Deep Learning parameters to evaluate the Deep Learning results with respect to (i) the accuracy of point placement and (ii) the accuracy of colour measurement. We then use the best identified parameter settings to assess the performance of the model when applied to point placement for the *Littorina* dataset. Using the inferred landmarks we present a simple biological application to compare *Littorina* shell shape in specimens from sites exposed to wave action or sites subject to crab predation. Finally, we discuss the potential for pose estimation approaches to replace manual labelling in similar phenotypic datasets.

## Materials and methods

### Identifying bird plumage regions

**a) Specimen imaging and annotation.** The images used in this study were taken in the bird collections at the Natural History Museum, Tring. All images follow a standardised set-up

**Fig 1. Examples of points on the avian specimen images.** (a) Five reflectance standards and five body regions (crown, nape, mantle, rump and tail) of the back view; (b) Three body regions (throat, breast and belly) of the belly view; (c) Two body regions (wing coverts and flight feathers) of the side view.

design as described in detail by Cooney *et al*. [14,15]. We briefly summarise the protocol here. Both visible light (i.e. RGB images) and ultraviolet light photos were taken from three views (back, belly and side) with specimens placed with heads on the left and tails on the right in the majority of images. The full dataset has 234,954 photos (39,159 specimens across 27 bird orders), with 117,477 photos for each of visible light and UV light. In some cases, specimens were positioned in non-standard ways due to the nature of the specimen (e.g. species with exceptionally long necks are often preserved with folded necks and heads lying across the belly).

We aim to identify 10 body regions across the three images as well as five reflectance standards (Fig 1). Bird body regions may be obscured in some images, and in such cases these regions were not labelled. We use the points to locate each body region so that, in downstream analyses, these can be used to measure the colour information of this region (e.g. using points as the centre point of a larger area).

We used a set of 5094 bird images from the full dataset, containing 1698 bird genera (with one specimen from one species per genus), encompassing 81% of all extant bird genera and 27 bird orders. Images are not evenly distributed by order (e.g. Passeriformes has more than 60% of the images and Falconiformes has less than 1% of the images) This is due to bird species being unevenly distributed among taxonomic orders, rather than bias in sampling. Our full dataset is also taxonomically imbalanced (the proportion of images per order of the full dataset and the 5094 genus-level dataset can be found in S1 Table). However, the 5094 genus-level dataset represents the full dataset well and captures much of the extent of variation in plumage colour, patterns, and body shape among birds.

To compare the effect of balanced versus imbalanced taxonomic sampling on Deep Learning performance, we further created two training sets (balanced and imbalanced) and two test sets (balanced and imbalanced). The balanced training set and the balanced test set have images evenly distributed in taxonomic orders, while the imbalanced training set and the imbalanced test set have a similar taxonomic distribution to the full dataset (the proportion of images per order of the imbalanced sets can be found in S1 Table). The balanced training set has 690 images with 30 images (10 specimens) per order. The balanced test set has 345 images with 15 images (5 specimens) per order. Here we only used 23 out of 27 orders, because four orders (Eurypygiformes, Leptosomiformes, Mesitornithiformes, Opisthocomiformess) that do not have 45 images in the full dataset were discarded. For the balanced training set, we sampled 30 images from the 13 orders that have at least 30 images in the 5094 genus-level dataset, and then we sampled images from the full dataset to make the remaining 10 orders reach 30 images (i.e. 213 images). For the balanced test set, 345 images were additionally selected from the full dataset.

The imbalanced training set and the imbalanced test set only have images from the 23 orders used in the balanced sets for better comparability. The imbalanced training set sampled

690 images from the 5094 genus-level dataset. While the imbalanced test set sampled 345 images that are not used in either the balanced training set or the imbalanced training set (i.e. the remaining 5094 genus-level images and images from the balanced test set).

In summary, 5652 images (5094 of the genus-level dataset, 213 of the balanced training set and 345 of the balanced test set) were labelled manually by one of the authors (YH) by placing points in the approximate centre of each body region and reflectance standard. These images were annotated using an annotation toolkit (https://github.com/EchanHe/PhenoLearn) developed by YH and a citizen science annotating web portal (https://www.zooniverse.org/projects/ghthomas/project-plumage/). These annotated images were used to train and evaluate the Deep Learning models.

**b) Deep learning pipeline overview.** Our Deep Learning pipeline involves three steps: (a) data pre-processing (Fig 2A), (b) model training (Fig 2B), and (c) model evaluation (Fig 2B). We used the Stacked Hourglass [33] method, which has been shown to perform exceptionally

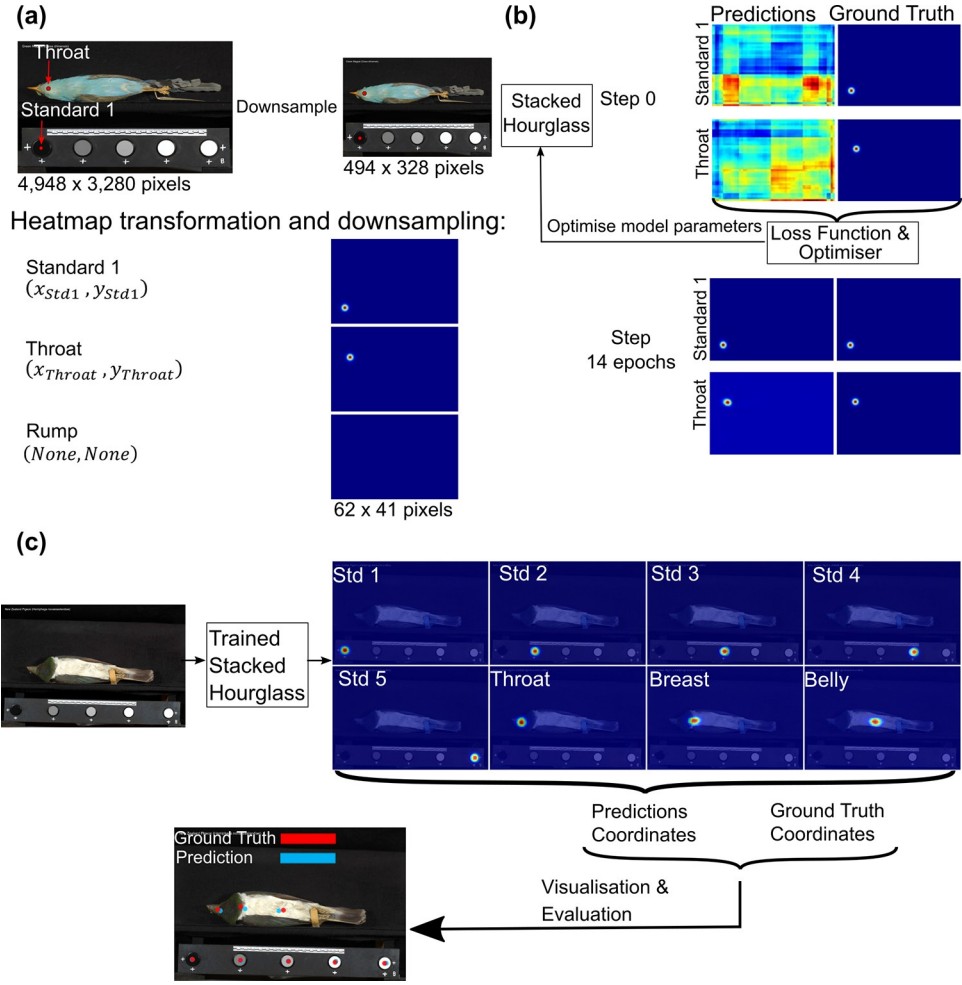

**Fig 2. The pipeline of applying the Stacked Hourglass model to predict points on the avian specimen dataset.** (a) A data pre-processing step resizes images (from 4948 x 3280 pixels to 494 x 328 pixels) and transforms point coordinates into heatmaps (62 x 41 pixels). (b) This pre-processed training data is then used to train the network. Output heatmaps iteratively become closer to ground truth during training. (c) The trained network is used to generate predictions of point locations for validation images. These are then post-processed (transforming heatmaps back to coordinates) and evaluated.

well on single-person pose estimation. The model is trained under pre-defined network hyper-parameters (i.e. configurations for the training process, such as training steps and learning rate). For each training step, the network generates predictions. The predictions are compared to the ground truth by calculating a loss function. The ground truth is defined as the ideal result in Deep Learning and in our analysis is the expert-labelled points. The network parameters are then updated (optimised) by applying gradient descent to minimise the loss function [44]. The network can therefore generate predictions that are iteratively closer to the ground truth (Fig 2B). When the network converges or training finishes, the expert-labelled (ground truth) data is used to evaluate the precision and accuracy of the validation set result from the trained network (Fig 2C).

**c) Data pre-processing.** We first converted all images to JPG from RAW format. In RGB format, each pixel has values ranging from 0 to 255 for each (red, green, and blue) channel. Due to the memory constraints of the graphics processing unit (here an NVIDIA GTX 1080Ti with 12GB GPU memory) and the model complexity, it was necessary to reduce the input resolution of the images. While the original application of the Stacked Hourglass model uses 256 x 256 pixels as the input resolution [33] the resolution of our raw images is 4948 x 3280 pixels. We down-sampled images 10-fold to 494 x 328 pixels using bilinear interpolation from the OpenCV computer vision library [45]. We conducted preliminary tests to ensure that this resolution could be trained on our GPU without any computational difficulties.

The Stacked hourglass model outputs probability heatmaps for the location of a point. Outputting heatmaps has been shown to outperform other output formats (e.g. point coordinates) in many pose estimation studies [33–35]. The CNN reduces the input resolution to extract image features based on convolution and pooling layers [25] resulting in heatmaps with a resolution of 62 x 41 pixels (eight times smaller than the input resolution). We used heatmaps generated from 2-dimensional Gaussian distributions (mean = 1, SD = 2) around each expert-labelled point as the ground truth (Fig 2A), which match the output of the Stacked Hourglass model. Where a body region did not appear in an image (i.e. it was completely occluded or did not occur in the view), an empty heatmap was used (Fig 2A). After pre-processing, the original data and labels were converted to input images with a resolution of 494 x 328 pixels and ground truth heatmaps with a resolution of 62 x 41 pixels.

**d) Model training.** Here, we used the 5094 genus-level dataset to train and evaluate the Stacked Hourglass model. We used five-fold cross-validation to evaluate the model performance, and the training and validation split is 80:20. To do this, we first split our data into five random and independent subsets (four sets with 1019 images, and one set with 1018 images). We then sequentially selected one subset as the validation set and combined the remaining four subsets as the training set. We therefore run five independent assessments of the model performance where each model is trained from the start and the trained models do not see the validation images. With this approach every image from the 5094 genus-level dataset is predicted and evaluated once. Cross-validation can provide an accurate estimate of model performance by averaging performance for different partitions (five partitions for five-fold cross-validation) of training and mutually exclusive validation sets.

Annotated images were divided into batches of four images, and one batch per training step was fed into the model to generate prediction heatmaps. Using batches balances the memory usage of the GPU and the optimisation of each step [46]. The loss function for the network, which shows the difference between predictions and ground truth, was calculated as the mean squared error between prediction heatmaps and ground truth heatmaps (the heatmap dimension: 62 x 41 x 15). To minimise the loss function, model parameters were updated using the ADAM optimiser [47] along with the gradient of the loss function (Fig 2B). The initial learning rate was set as 0.01. Through the training process, the learning rate was cosine decayed and

restarted at the initial value after reaching zero, which increases the possibility of reaching a better local optimum [48]. The length of the first period of decay-restart was set to one epoch, where an epoch is defined as one pass of the full training set for the network. After each period, the new period was two times longer than the previous one (i.e. the second period takes two epochs to decay to zero, the third period takes four epochs and so on). The model was trained over 15 epochs for four complete decay-restart periods, after which the model converged (i.e. the loss stopped decreasing). The model (https://github.com/EchanHe/Pose_DL_specimen_pts) was implemented using Python 3 and Tensorflow 1.12 Deep Learning library [49] on one NVIDIA GTX 1080Ti GPU (12GB GPU memory).

**e) Model evaluation.** After training, validation images were fed into the trained network to generate prediction heatmaps. Heatmaps were used to generate the coordinates of the predicted points. To generate points that match the input image resolution (4948 x 3280 pixels), heatmaps were resized back to the original resolution. Predicted coordinates for each point were generated using the location of the maximum value (or the average of locations if there are multiple maximum values) of the corresponding heatmap (Fig 2C). The network generates 15 points for each image and points that are not represented in a particular view (e.g. the 'crown' point in ventral images) were discarded in the evaluation. To make predictions and ground truth comparable, occluded points in the expert-labelled dataset were not evaluated.

We evaluated model performance at the original image resolution (4948 x 3280 pixels) and used metrics that describe the accuracy of predictions of the validation set from both geometric and colour perspectives. The pixel distance is the most straightforward metric for comparing geometric accuracy and is simply the Euclidean distance (measured in pixels) from the input ground truth point coordinates to the predicted coordinates (i.e. the pixel location with the maximum heatmap value). Pixel distance can be used to assess each point individually and as the average for all points within an image.

An alternative geometric measure commonly used in pose estimation is the Percentage of Correct Keypoints (PCK). PCK is the percentage of predictions that have pixel distances below a given threshold (e.g. in human pose estimation studies this is typically human head length) [36]. Birds from our dataset can be very variable in size and shape (e.g. penguins vs. humming-birds), it is difficult to conceive of a single dynamic threshold equivalent to human head length. Therefore, here we used a fixed threshold of 100 pixels (PCK-100) instead. PCK-100 is suitable for evaluating the accuracy of the five reflectance standard predictions because the minimum radius of reflectance standard circles is slightly greater than 100 pixels and ground truth points of reflectance standards were always placed in the centre of standards. As a result, the PCK-100 metric is most useful for measuring the accuracy of the predicted coordinates of the five reflectance standards in each image.

For some applications, point accuracy may be critical (e.g. placing landmark points for geometric morphometric analyses–see *Littorina* data below). However, our primary goal of placing points on bird specimens is to extract colour information for regions. To assess the extent to which human and predicted point coordinates result in similar estimates of colouration, it is therefore necessary to evaluate similarity in the colour information between model-predicted and expert-labelled ground truth point. We used two methods to extract colour information: Bbox-20 and Heatmap-90. For Bbox-20, we defined an area around the body region point using a 20 x 20 bounding box, mirroring previous manual approaches [15], and we used pixels inside the box to calculate the colour information. A 20 x 20 bounding box is small enough to account for the smallest body region within the dataset. However, this method does not account for variation in region size (e.g. the rump is usually larger than the crown). We therefore used another extraction method that takes account of size variation. Stacked Hourglass outputs probability heatmaps and they can capture size variation of the focal body region.

For example, S1 Fig shows prediction heatmaps of the crown and rump on an image, where the rump (S1B Fig) has a larger heatmap than the crown (S1A Fig). To use heatmaps in colour extraction, we first interpolated heatmap values to the range 0 to 100, where a value corresponds to how likely the point is located in that pixel. We used 90 as the threshold value to determine which pixels should be included within each body region and we refer to this extraction method as Heatmap-90. The averaged RGB values of pixels selected by Bbox-20 and Heatmap-90 were used as the colour information, and we calculated the correlation coefficient (Pearson's correlation coefficient R) between ground truth and predictions in order to measure colour accuracy. Taken together, pixel distance, PCK-100 and colour correlation capture the precision, accuracy, and biological relevance (i.e. plumage colour measurement) of the predicted points.

We manually checked predictions and classified each image as 'correctly' or 'incorrectly' predicted. Specifically, for a given image if all points are placed somewhere inside the correct corresponding body regions, we considered the image to have been correctly predicted. However, if at least one point was placed outside its corresponding body region, the image was considered to have been incorrectly predicted. Body regions are visualised virtually on specimens by experts based on their anatomical knowledge. The boundary of a body region can be subjective, and we want to minimise the human variance and error, so predictions were cross-checked by two people (YH and CRC).

We further evaluated the Deep Learning result by taking variation among human labellers into account. Humans may place points differently on the same image, and if the difference (i.e. error) among humans is similar to the difference between human and Deep Learning, then it is possible to say that the predicted results are sufficiently accurate as to be indistinguishable from the difference in accuracy expected between human labellers. To test this, we randomly selected 300 images (100 images per view) from the 5094 genus-level dataset to be re-labelled by two other experts (CRC and GHT). The pixel distances between the original expert points (YH) and points from the expert non-trainers (CRC and GHT) were used to quantify human variability. Then, pixel distances were calculated pairwise from four results (YH, CRC, GHT and predictions), which can be categorised into three groups by datasets used for comparing (i) predictions vs expert trainer (i.e. between predictions and YH), (ii) predictions vs expert non-trainers (i.e. between predictions and CRC or GHT), (iii) between experts. We then compared the difference among these three groups.

**f) Experimental runs with alternative pose estimation models and image manipulations.**   In addition to the core pipeline above, we explored possible improvements to the approach by manipulating the model in various ways: (i) using an alternative pose estimation model, Convolutional Pose Machine (CPM) [(35)]; and (ii) using alternative input resolutions (329 x 218 and 247 x 164). Furthermore, we tested the effect of using low-quality images as the training set, where we applied some image transformations on the images to make four low-quality datasets. The low-quality datasets are: (i) rotation (angles between -45˚ to 45˚), (ii) translation on both x and y axes (-500 to 500 pixels), (iii) horizontal flip 50% images randomly, (iv) the combination of all three transformations. The details of these runs and results can be found in the S1 Appendix. In the main text, we focus on results based on the core pipeline since this proved to be the best model configuration in our evaluation tests.

**g) The effect of taxonomic imbalance on the model performance.**   The full dataset is taxonomically imbalanced, we thus conducted a further experiment to explore whether this imbalance affects the model performance. We used two training sets and two test sets described previously, the balanced training set (690 images with 30 images per order), the imbalanced training set (690 images and has a similar taxonomic sampling to the full dataset.

See S1 Table), the balanced test set (345 images with 15 images per order) and the imbalanced test set (345 images and has a similar taxonomic sampling to the full dataset. See S1 Table)

We then compared the model performance (evaluated by the pixel distances of the body region predictions) for the two test sets from models trained by the two training sets. We did not include the five reflectance standards in the evaluation, as we only wanted to focus on predictions on birds. To further explore the effect caused by taxonomic imbalance, we train models by applying leave-one-order-out on the balanced training set (i.e. 23 models were trained), where we trained a model using the balanced training set without images from an order each time and evaluated the predictions on the balanced test set. All the models described in this section were trained under the same configuration as described in the model training section.

## Placing morphological landmarks on *Littorina* shells

**a) Specimen imaging and annotation.** We further tested the performance of the Deep Learning pipeline on a dataset that requires highly accurate point placement: placing morphological landmarks on *Littorina* shell images. These landmarks are used to quantify shell shape, and have been used to test, for example, the prediction that *Littorina* shell morphology reflects ecological differences in habitat (crab-rich or wave-swept intertidal habitats) [50,51]. These analyses help to explore the possible wider utility of this pipeline for processing other biodiversity datasets, as well other use cases. Here, we used 1410 images as the expert-labelled dataset for the Deep Learning pipeline.

In each image, the shell was placed on a pink colour Play-Doh background, next to 5-mm graph paper, with a label of the shell ID on the side. All shells were imaged under a Leica M80 (8:1 zoom) microscope using a Leica MC 170 HD camera. The digitised images have different pixel resolutions (1280 x 960, 1944 x 1458, 2074 x 1556, 2592 x 1944, 5760 x 3840). We used Deep Learning to predict seven landmarks (Fig 3), which are part of a 15 landmarks morphometric configuration introduced in Ravinet et al. [50]. We focused on these seven landmarks as they are the landmarks that can be place directly and are used to infer the rest eight landmarks. 1410 images were annotated using tpsDig2 [52].

**b) Deep learning.** We applied the Deep Learning pipeline with the optimised model parameters derived from the bird colour analyses to the *Littorina* data. While the parameters could potentially be fine-tuned further, our approach serves to demonstrate the broad applicability of a single pipeline to diverse labelling challenges.

Images were first resized into a uniform resolution using bilinear image interpolation in OpenCV [45], as well as coordinates of landmarks. The resolution of 2592 x 1944 was used because the majority of images have this resolution. We again used five-fold cross-validation and split 1410 annotated images into training and validation sets with an 80:20 ratio. We used the Stacked Hourglass model to train the dataset with an input resolution scale of five (518 x 388 pixels), a learning rate of 0.01, a batch size of four, and a training length of 15 epochs. The learning rate was cosine decayed and restarted at the initial value after reaching zero. For model evaluation, we used pixel distance and PCK to evaluate the quality of predictions on the validation images. We used 5% of the shell length (The distance between LM1 and LM7, red dotted line in Fig 3) as the dynamic threshold for PCK (PCK-L5). All coordinates were evaluated under the resolution of 2592 x 1944.

**c) Testing for shape differences between ecotypes.** We further evaluated the accuracy of extracted phenotypic information using 188 images of the expert-labelled images with available ecotype information. Shell is classified as one of two ecotypes: crab (N = 100) or wave (N = 88). We tested whether the landmarks placed by our Deep Learning pipeline can recapitulate the separation between crab- and wave-type shell shapes observed in previous studies

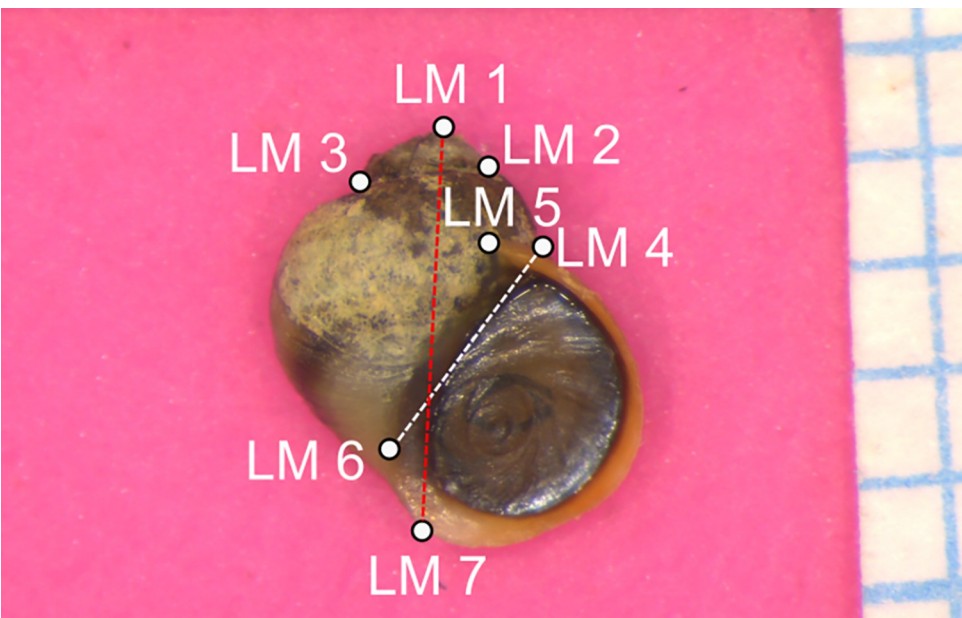

**Fig 3. An example of a *Littorina* shell image and seven landmarks.** LM1 is defined as the apex of the shell; LM2 is defined as the upper suture of the penultimate whorl (right); LM3 is defined as the upper suture of the penultimate whorl (left); LM5 is defined as the end of the suture; LM6 is defined as the point at the external edge of the lip and the line between LM4 and LM6 (white dotted line) is tangent to the operculum; LM7 is defined as the point at the bottom of the shell and the line between LM1 and LM7 (red dotted line) is tangent to the aperture.

based on expert landmarks [50,51]. To do this, we first ran Generalized Procrustes Analysis (GPA) to align landmarks using the R package Geomorph [53]. We then applied Principal component analysis (PCA) on aligned landmark coordinates to summarise the shape variation. GPA and PCA were applied to landmarks of the ground truth and predicted results (a total of 2632 landmarks from two sets of landmarks). We used MANOVA to measure if ecotypes and labelling methods (i.e. experts or the Stacked Hourglass) affect the shape variation. To quantify the similarity between shapes measured by landmarks from two methods, we calculated the group distances among four groups (between centroids) that have different combinations of ecotypes and labelling methods using the R package dispRity [54]. The four groups are: (i) crab ecotype with ground truth landmarks; (ii) wave ecotype with ground truth landmarks; (iii) crab ecotype with Deep Learning landmarks; (iv) wave ecotype with Deep Learning landmarks.

## Results

### Stacked Hourglass on avian specimen images

For the avian specimen dataset, the pixel distance between expert and predicted points averaged 47.3 (1.4% of the image height– 3280 pixels) across all 15 points (10 body regions and 5 reflectance standards). Reflectance standards (average pixel distance: 21.9) were predicted more accurately than body regions (average pixel distance: 86.1) (Fig 4A). All reflectance standards scored 100% for the PCK-100 metric (Fig 4B), suggesting all standard locations were correctly predicted as they were located inside the circular reflectance standard. As mentioned, the model was less reliable for body regions, most notably the tail, rump, flight feathers, and coverts, which have relatively large pixel distances and high variance (Fig 4A). These results might be influenced by these regions sometimes not being present in the training images

(a)

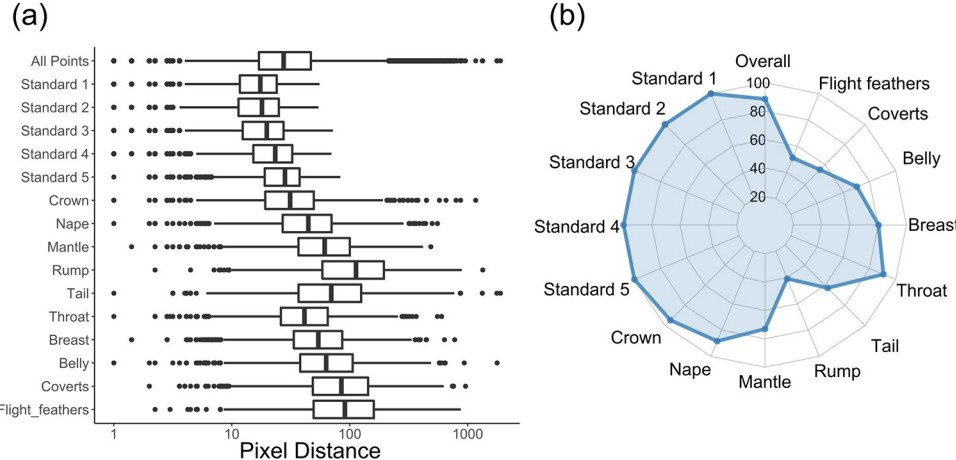

(c)

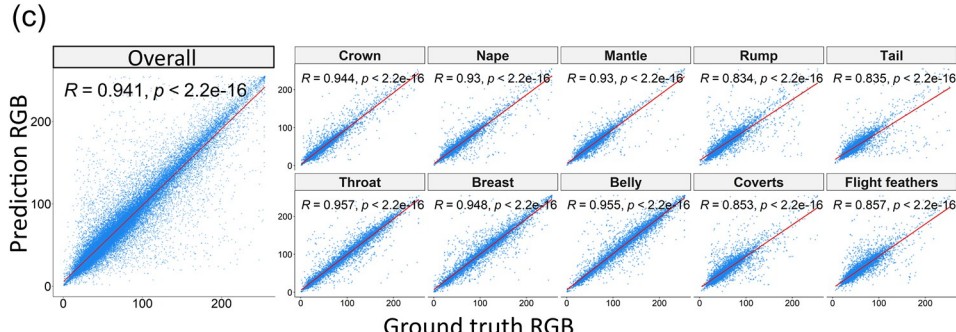

**Fig 4. Evaluation results between ground truth and predictions for the avian specimen dataset.** (a) Pixel distances. The x axis (pixel distance) is logarithmic scaled; (b) PCK-100; (c) RGB Colour correlations (colour extraction method: Heatmap-90).

(rumps were frequently obscured or partially occluded–e.g. Fig 5C) or being small and indistinct to the human eye (e.g. Fig 5H). Although body regions were less accurately predicted than standards, the worst region (rump) has an average pixel distance of 147.4 which is approximately 4.5% of the image height and a PCK-100 of 44.6%. Fig 5 shows some correct and incorrect predictions with their explanations in the figure legend.

The overall colour correlation coefficient between expert and machine-derived RGB values is higher than 0.91 for both extraction methods (Bbox-20: 0.914 and Heatmap-90: 0.941). RGB values extracted by the Heatmap-90 approach have higher correlation coefficients than those using the Bbox-20 approach (S2 Table), and therefore we focus on correlation results based on Heatmap-90. The four body regions with the lowest colour correlation coefficients are the tail, rump, flight feathers, and coverts (Fig 4C), as expected based on pixel distance (see above). However, even the region with the lowest correlation coefficient (rump) has a relatively high correlation value (R = 0.834). This suggests that the Deep Learning algorithm provides accurate estimates of colour suitable for downstream analysis. The detail of the scores (i.e. pixel distance, PCK-100, colour correlation) can be found in S2 Table.

We then manually checked the predictions, which provides further verification of how well the Deep Learning model can predict body regions on bird specimen images. In total, predictions on 234 images (back view: 97, belly view: 15 and side view: 122) out of 5094 were considered incorrect after expert assessment. Conversely, more than 95% of the images were classified as correctly predicted. In addition, we compared incorrectly predicted images

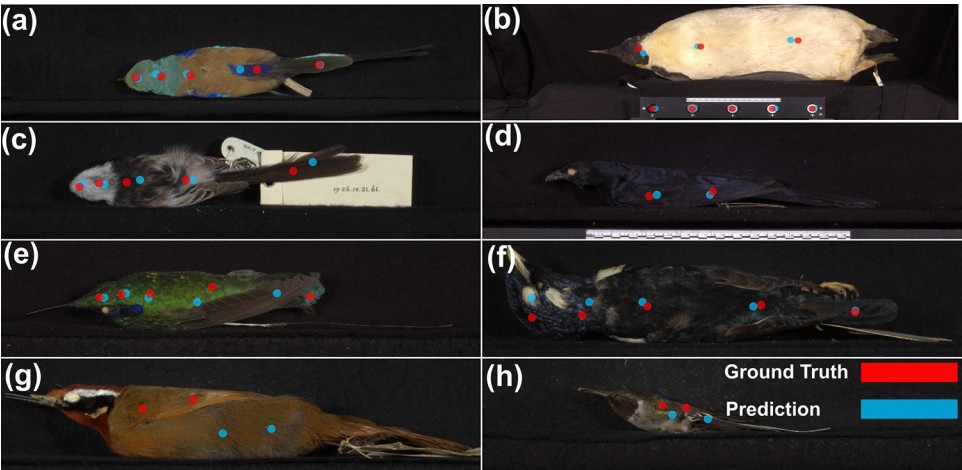

**Fig 5. Example images of ground truth and predictions for the avian specimen dataset.** Images (a)-(d) are correct predictions: (a) a specimen with high colour diversity, (b) a big specimen and very small reflectance standards, (c) a specimen with interferential objects such as a specimen tag, (d) a specimen which is similar to the background. Images (e)-(h) are incorrect predictions: (e) The tail is partially occluded by a wing, and it was incorrectly labelled as the wing. (f) The eye of the bird was misidentified as the crown, while the specimen was placed in a rare posture (placed on its back rather than belly) in the image. (g) The predictions of the wing were placed on the body, and the wing has a similar colour to the rest of the body regions. (h) The predictions were not placed on the wing which is very small. (Note: we cropped only focal parts of images to achieve better visualisation).

(n = 234) with images that one expert (YH) flagged as 'difficult to label' (n = 135) without knowing the predictions, and there are 32 overlapped images. The causes of the labelling difficulties are (i) small body region, (ii) similar colour to the adjacent area, (iii) rare posture of the specimen, and (iv) partially occluded body region. The detail of the human error checking can be found in the S1 Appendix and S2 Fig.

We further assessed variations in pixel distances across three groups to compare among predictions, the expert trainer (YH), and expert non-trainers (GHT, CRC). The pixel distance comparing predictions to the expert trainer was not significantly different from the pixel distance comparing between other experts (Fig 6), suggesting that pose estimation can perform as well as expert-labelling on this dataset. For individual regions, predictions vs trainer performed better than between experts in five regions (nape, mantle, breast, belly and coverts). The overall pixel distance from predictions to non-trainers, however, was greater than the other two groups, suggesting that the model does not predict labels from non-trainer as reliably. Pixel distances on the five reflectance standards between experts were all significantly smaller than ones from predictions vs expert trainer (mean differences range: 2.9–13.2 pixels).

## The effect of taxonomic imbalance on the model performance

The average pixel distance of the predictions for the balanced test set predicted using the balanced training set is 134.9 (2.6% of the image width). This compares with an average pixel distance of 172.6 (3.4% of the image width) for the imbalanced training set. The average pixel distance of the predictions for the imbalanced test set predicted using the balanced training set is 107.2 (2.1% of the image width). This compares with an average pixel distance of 110.9 (2.2% of the image width) for the imbalanced training set. The average pixel distances for each order are listed in S3 Table. For predictions on the balanced test set, the pixel distances predicted from the balanced training set were significantly smaller than the ones predicted from the imbalanced training set (t(2279.5) = -5.12; p < 0.05). Pixel distances from the balanced

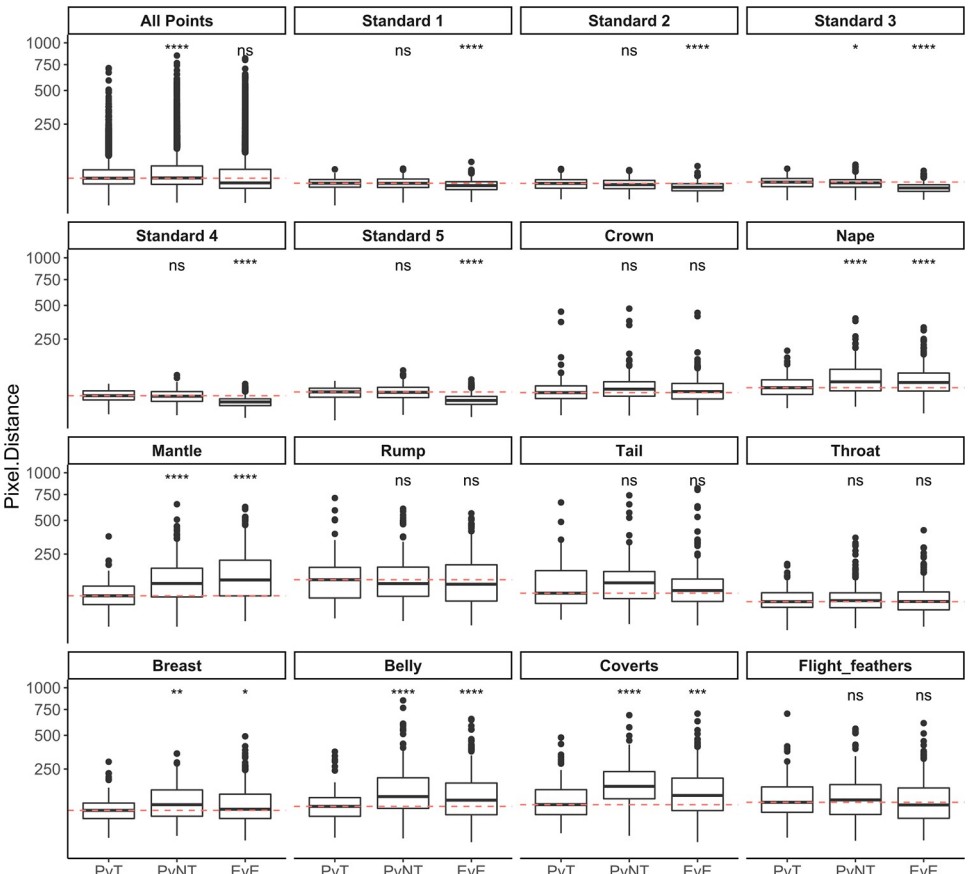

**Fig 6. Pixel distances for experts and Deep Learning comparison.** Pixel distances are compared across three groups: predictions vs expert trainer (PvT); predictions vs expert non-trainer (PvNT); between experts (EvE). Red dotted lines are the median of PvT. Significant symbols are t-test that compares PvT against PvNT or EvE (ns: $p > 0.05$; *: $p < = 0.05$; **: $p < = 0.01$; ***: $p < = 0.001$; ****: $p < = 0.0001$). All Y axes (pixel distance) are square root scaled.

training set were significantly smaller for five out of 23 orders (Fig 7A and S3 Table), while no significant differences for the remaining 17 orders. For predictions on the imbalanced test set, the pixel distances predicted from the two training sets were not significantly different (t (2155.9) = -0.72; p = 0.77). When comparing order-wise results, using the imbalanced training set produced significantly better results (t(1354.4) = 2.88; p < 0.05) for the Passeriformes (more than 60% of the images in the imbalanced test set) than using the balanced training set (Fig 7B and S3 Table). The balanced training set achieved a better result for the Bucerotiformes (Fig 7B and S3 Table), but the imbalanced test set only has three images from this order (S1 Table). For the leave-one-order-out tests (evaluated on the balanced test set), two orders (Coliiformes and Galliformes) were predicted significantly worse when using training sets without them (S4 Table).

Taken together, the balanced training set outperformed the imbalanced training set for predicting the balanced test set. However, the predictions on the imbalanced test set were not affected by taxonomic sampling in training sets. Therefore, creating a training set that is representative of the full data would be good practice for either taxonomically balanced or imbalanced datasets. The accuracy of a taxonomic group could be limited if images of that group are not well-represented in the training set.

(a)

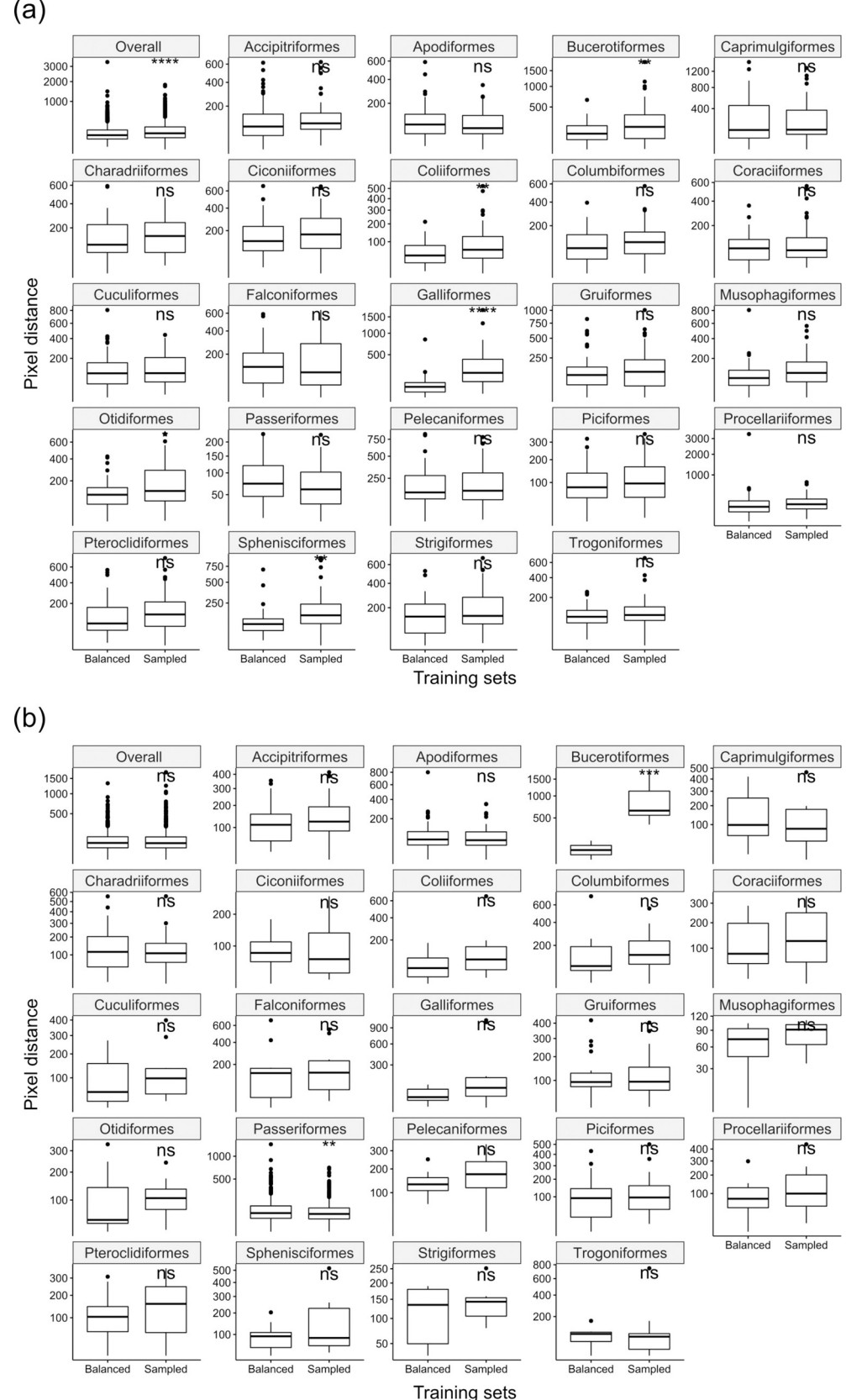

(b)

**Fig 7. Pixel distances for balance and imbalance in taxonomic groups comparison.** (a) Comparing the performance between using the balanced training set and using the imbalanced training set on the balanced test set. (b) Comparing the performance between using the balanced training set and using the imbalanced training set on the imbalanced test set. The performance was evaluated on the pixel distance. Significant symbols are t-test results (ns: p > 0.05; *: p < = 0.05; **: p < = 0.01; ***: p < = 0.001; ****: p < = 0.0001). All Y axes (pixel distance) are square root scaled.

## Experimental runs results

Stacked Hourglass was clearly a better architecture than CPM (predicted points from the Stacked Hourglass were on average 29.5 pixels lower than the points from CPM). We also found that the model with 494 x 328 pixels (the highest resolution we tested) achieved the best result when compared to lower resolutions (329 x 218 and 247 x 164). Pixel distance (from Stacked Hourglass) using 494 x 328 input resolution is 4.1 pixels better than one using 329 x 218 and 36.0 pixels better than one using 247 x 164. The detail of these results can be found in the S1 Appendix, S3 Fig and S5 Table.

We found that low-quality datasets had a significant negative effect on accuracy (ANOVA of pixel distance for all points: F = 87.2; df = 4.0, 210,700; p<0.01. S6 Table shows ANOVA results for individual points). Performances of different transformed datasets were consistently worse than the original dataset as shown in S4 Fig. However, the overall average pixel distance differences between the original dataset and low-quality datasets was less than 10 pixels (Dataset (i): 4.8, Dataset (ii): 3.6, Dataset (iii): 9.1, Dataset (iv): 8.1), which were more accurate than using CPM suggesting that the effect of low-quality data on performance is minor.

## Placing morphological landmarks on *Littorina* shells

We applied the Stacked Hourglass model to infer seven morphological landmarks on a dataset of *Littorina* shells. The evaluation results are listed in Table 1. The average pixel distance across all seven landmarks is 18.9 pixels, which is less than 1% of the image height (1944 pixels). The average PCK-L5 of 95.8% indicates more than 95% of the predicted landmarks were placed within 5% of the shell length. The least accurate (both pixel distance and PCK-L5) landmark is LM6. Compared to other landmarks that are either on the shell outline (LM1,2,3,4,7) or at a well-defined location (LM5), the lack of distinct features of LM6 could limit the accuracy (i.e. difficult for the machine to learn the landmarking pattern).

We estimated morphospaces for the 188 shells with ecotype information. The first two PC axes based on the seven landmarks explain 62.9% of the total shape variation (Fig 8), and the third to eighth PC axes explain 32.4% (S5 Fig). We found a clear separation between crab and wave on PC1 for both ground truth and Deep Learning landmarks (Fig 8), where wave specimens tend to have large PC1 values and crab specimens have small PC1 values. PC values

**Table 1. The pixel distance and PCK-L5 for Littorina shells.**

| Landmark | Pixel Distance | PCK-L5 |
|---|---|---|
| All LMs (N = 9870) | 18.9 | 95.8 |
| LM1 (N = 1410) | 11.6 | 99.8 |
| LM2 (N = 1410) | 12.0 | 99.6 |
| LM3 (N = 1410) | 13.5 | 98.9 |
| LM4 (N = 1410) | 13.9 | 99.5 |
| LM5 (N = 1410) | 19.9 | 97.4 |
| LM6 (N = 1410) | 43.7 | 77.4 |
| LM7 (N = 1410) | 18.3 | 97.7 |

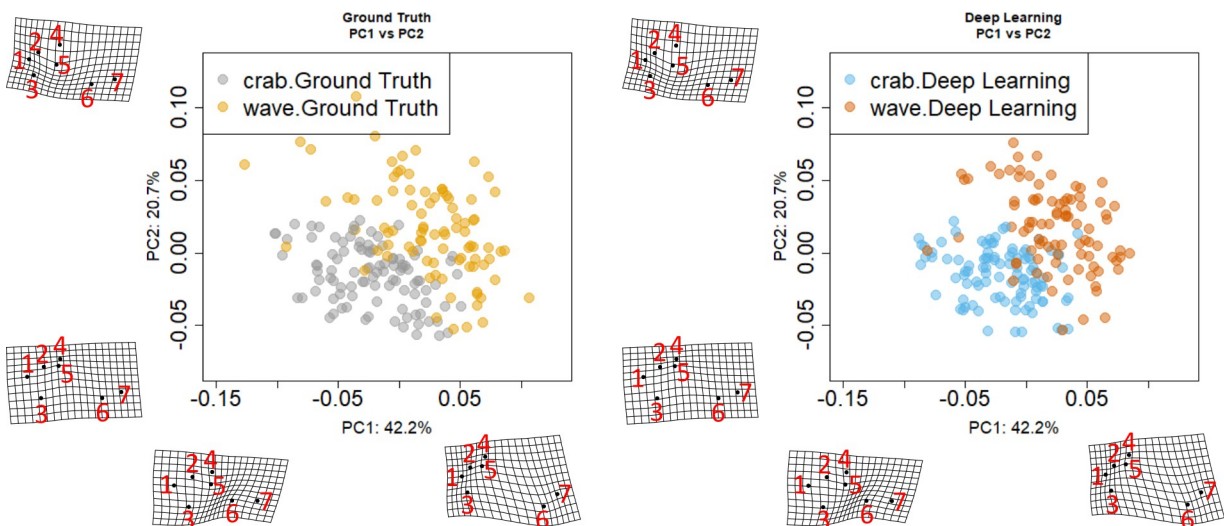

**Fig 8. Distributions of PC1 and 2 for *Littorina* shells.** PC1 and PC2 explains 62.9% of the total variation. Left: the ground truth landmarks (Crab: Grey; Wave: Yellow). Right: Deep Learning predicted landmarks (Crab: Blue; Wave: Red).

between ground truth and Deep Learning are all positively correlated for the first eight axes (S6 Fig), notably, the correlations for PC1 (R = 0.89) and PC2 (R = 0.9) are very high. The result of MANOVA (S7 Table) shows that the ecotype contributes to the shape variation (p<0.001), but the labelling methods (i.e. experts or the Stacked Hourglass) did not (p = 0.95), where we used PC1-8 as the shape variation. Similarly, we found that centroid distances between groups with the same ecotype but different labelling methods are smaller than distances between groups with different ecotypes but the same labelling method (S8 Table), suggesting that the landmarks placed by the Stacked Hourglass can extract shape information for *Littorina* shells that is highly similar to expert-placed landmarks.

## Discussion

We found that pose estimation methods can automatically and accurately locate points on two distinct datasets that present distinct labelling challenges. For the avian dataset, we tested whether pose estimation networks could be used to identify points to extract colour information from regions. Pose estimation methods were originally designed to identify human body parts and joints in human posture photos [33,35,36,55]. Human posture photos often contain varieties of human postures, clothes, and backgrounds, and, from the human perspective, seem more visually complex than the specimen photos taken in a consistent digitisation setup (e.g. same orientation, background, lighting). However, while posture can create clearly defined points, and the *Littorina* data are broadly similar in that regard, the focal regions we aim to identify in the avian dataset are more ambiguous. This creates a different set of challenges for expert labelling and Deep Learning alike that could potentially limit the performance of Deep Learning algorithms. However, our analysis suggests that Deep Learning can perform remarkably well.

To evaluate results we used standard geometric metrics (i.e. pixel distances and PCK) that are commonly used in evaluating pose estimation studies [33,35,36]. These are well suited to evaluating point placement where precision is critical, as is necessary for landmarking in geometric morphometrics. Indeed, for the *Littorina* dataset, which has precisely-defined landmarks and is similar to pose estimation tasks, we found that more than 95% of the predicted

landmarks were placed within 5% of the individual shell length. We also showed that these predicted landmarks generate shape estimates that are highly similar to those derived from human-placed landmarks, and that these machine-derived shape variables can accurately capture established differences between *Littorina* shell ecotypes.

We used two additional metrics (colour correlation and experts' evaluation) when evaluating the quality of predictions for the colour extraction of birds. This evaluation is important because while some applications will require very high accuracy of points, others require only that the point placement is sufficient to extra relevant data for downstream analysis. The colour correlation coefficients reflect the accuracy of colour measurements extracted based on Deep Learning predictions. An overall colour correlation coefficient of 0.941 and per-region correlations that are greater than 0.83 show that Stacked Hourglass can reliably place points and extract accurate colour information that is highly correlated with colour derived from expert-labelled points. By comparing the extracted plumage colour and morphospaces, we are able to show that the Stacked Hourglass generates accurate and biologically meaningful measurements, as geometrical metrics (e.g. pixel distance) may not reflect the biological accuracy well. For example, a point in the plumage and a point in the background might have a small pixel distance, but the colour information they extracted is completely different.

However, it is clear that even with overall high accuracy, researchers should scrutinize Deep Learning predictions to minimise the risk of perpetuating errors in downstream analyses. Checking all the predictions is feasible for small datasets, but it is impractical for large datasets, which are common for natural history datasets. Automatically selecting images that are more likely to have incorrect predictions is a possible solution to speed up the reviewing process. Statistical or heuristic methods can be used to flag possible error predictions. For example, (1) the predictions of the reflectance standards can be flagged if their pixel values are different from the expected values (e.g. a prediction of the leftmost standard should have a pixel value close to black) or (2) an image can be flagged if it is an outlier in the morphospace. In addition, we can prioritize checking images with error-prone traits (e.g. if specimens from a certain taxonomic group are likely to have incorrect predictions) that are found in the validation stage. Taken together, using these methods can substantially increase accuracy while spending limited time, allowing researchers to find an optimal spot on the accuracy-time tradeoff.

Body regions are not truly homologous parts (e.g. landmarks in morphology studies), and there are often subtly different opinions regarding the area demarcating different body regions. As a result, even human labellers will often place body region labels in slightly different locations. In our analysis, variation among human labellers was calculated based on annotations of three observers with expert knowledge of bird anatomy and revealed that variation among experts is similar to the error in Deep Learning predictions. This offers reassurance that Deep Learning is a viable approach. However, our results also point to the value of including multiple labellers. The predictions are, as expected, closer to the trainer's points than to the points placed by non-trainers. Thus, for this task at least (that is, locating body regions on images of avian specimens), no single expert can be considered to be the absolute 'gold standard' and including points from multiple experts in building the training data would minimise bias towards the interpretation of a single expert and potentially improve the overall accuracy of predicted data.

As the avian dataset covers a wide taxonomical range, creating training sets that account for the variation introduced by the imbalance among taxonomic groups could affect the Deep Learning performance. Sampling a training set that is under a similar distribution of the full dataset is common in many Deep Learning use cases. Our result shows that using a balanced training set (images that are evenly sampled from each taxonomic group) performed well for both taxonomically balanced and imbalanced test sets. However, as some taxonomic groups

may not have sufficient images, creating a balanced training set from an imbalanced dataset can make the training set small which could limit the model performance. For the imbalanced data which our full dataset also is, our results show that using a balanced or imbalanced training set does not make a significant difference. Therefore using an imbalanced and representative training set can maximize the training set size while maintaining accuracy. Taken together, creating a training set that is representative of the full data could again be an optimal solution for datasets that are either taxonomically balanced or imbalanced. In the leave-one-order-out test, the model performed significantly worse on two orders if images from these orders were not in the training set, suggesting missing images from a taxonomic group in the training set could limit the model performance on that group. The remaining 21 orders were not significantly worse, which may be due to the small test size (15 images per order) or that the model learns the features of missing orders from other orders. Nonetheless, while our results suggest that average performance can remain high, there remains the possibility that predictions for groups that are underrepresented in the training set can be inadequate. This highlights the need for researchers to pay extra attention to groups that are badly underrepresented or absent from the training set. If some groups are poorly represented in the training set then it might be necessary to exclude them completely from the dataset for Deep Learning and to label them manually. Another possible solution, if manual labelling is not practical, is to use data augmentation (e.g. rotate, zoom and pan the original image) to create more training images for the underrepresented groups.

If new images became available, and these were taken under the same conditions as our existing data, we would expect the trained model to identify body regions accurately if are from groups that are already well-represent in the training set (e.g. Passeriformes). However, the trained model would be expected to make less accurate predictions on underrepresented groups (e.g. Galliformes). If the new images are mainly from underrepresented groups and there are enough images to make a new training set, we recommend to re-train the model. In addition, to generalise our method to new datasets (e.g. the *Littorina* dataset in this paper) that have different species or different labels, re-training the model with a training set from the focal dataset is necessary. Based on our results, we propose a general workflow to label points on other large-scale biodiversity datasets, with steps and recommendations listed in Table 2.

**Table 2. Recommendations on each step of building Deep Learning pipelines to label points on large-scale biodiversity datasets.**

| | Steps | Recommendations |
|---|---|---|
| 1 | Digitisation | • Use a consistent digitisation setup (e.g. background, lighting) where possible.<br>• Do not be overly concerned about the setup, as previous studies and the result on using low-quality datasets have shown to make accurate predictions on datasets with less consistent setups. |
| 2 | Labelling and creating a training set | • The training set should be labelled at least by one expert.<br>• Have multiple experts label and check the dataset to maximise its quality.<br>• Images used in labelling should be representative of the whole dataset. If the dataset has many taxonomic groups, try to include images from all groups if possible. Data augment or exclude the images if they are from badly underrepresented groups. |
| 3 | Training the model | • Train a model using its original configuration. Then try to train models with different configurations which are practical and easy for researchers to manipulate. |
| 4 | Evaluation | • Use appropriate metrics to evaluate the predictions. For example, use colour correlation to assess the colour information accuracy. |
| 5 | Post-processing | • If the dataset is small, check predictions manually to increase the accuracy.<br>• If the dataset is large, use statistical methods to flag images or check predictions with high error risk (e.g. taxonomic groups with high error rates). |

Overall, our results show that pose estimation can reliably identify points of interest on two distinct biodiversity datasets. It not only performed well on points with precise definitions (landmarks on *Littorina* photos) but also performed well in locating less precisely defined locations (body regions on avian specimen images). The latter is an important observation because it is different from the original objective of pose estimation methods. The model also performed well on avian specimen images with less consistent setups (i.e. the low-quality datasets), suggesting it has the potential to identify points on other natural history datasets, where some datasets may not be digitised under a high standard. The major advantage of applying Deep Learning is that it can significantly increase the speed of data measurements. The avian specimen dataset is a subset of a much larger set of 234,954 images that would take years to annotate by experts. A test run suggests that these can be annotated in three days. As hardware (e.g. GPU) and Deep Learning techniques are developing rapidly, using Deep Learning to make high-throughput and accurate phenotypic measurements on digitised natural history collections may become the mainstream, supporting more accurate predictions, more annotation types (e.g. segmentation), more data modalities (e.g. videos and sounds). It will be critical for researchers to either assess the performance of Deep Learning on their own data types or to error check afterwards, rather than simply treat Deep Learning as a black box. Nonetheless, the approach appears to offer considerable promise that complements large scale digitisation initiatives in driving the mobilisation of natural history data.

## Supporting information

**S1 Appendix. The detail of methods and results for the experimental runs and the detail of results for the human error checking.**
(DOCX)

**S1 Fig. Predicted heatmaps.** Heatmaps are from two body regions on a representative bird specimen image. (a) The predicted heatmap of the crown has a smaller area than the one of the (b) rump which has an ellipse-like shape and can capture more area of the rump region than using a fixed-size area.
(TIF)

**S2 Fig. Histograms of incorrect predictions flagged by experts.** Error prediction counts (N = 308) for each body region.
(TIF)

**S3 Fig. Results with different network architectures and input resolutions for the avian specimen dataset.** The plots show comparisons of model performance by comparing metrics from the ground truth data with the model prediction: (a) Pixel distances. The Y axis (pixel distance) is square root scaled; (b), PCK-100 of all and individual points; (c) RGB colour correlation coefficients of all and individual point-defined body regions (colour extraction method: Bbox-20).
(TIF)

**S4 Fig. Pixel distances of the original and four low-quality datasets.** The average pixel distance (the top left plot) of the original dataset is significantly smaller than the pixels distances of the low-quality datasets (ns: $p > 0.05$; *: $p < = 0.05$; **: $p < = 0.01$; ***: $p < = 0.001$; ****: $p < = 0.0001$). The four low-quality datasets are: (i) rotation (angles between -45˚ to 45˚), (ii) translation on both x and y axes (-500 to 500 pixels), (iii) horizontal flip 50% images randomly, (iv) the combination of all three transformations. The Y axes (pixel distance) are square root scaled.
(TIF)

**S5 Fig. Distributions of PC3-4, PC5-6 and PC7-8 for *Littorina* shells.** PC3-8 explain 32.4% of the total variation. Left: the ground truth landmarks (Crab: Grey; Wave: Yellow). Right: Deep Learning predicted landmarks (Crab: Blue; Wave: Red).
(TIF)

**S6 Fig. Correlations of PC1-8 between ground truth (X axes, N = 188) and Deep Learning (Y axes, N = 188) of shell shape.**
(TIF)

**S1 Table. Tables of images counts and proportions for each order in four datasets.** The datasets are (a) the full dataset; (b) the genus-level dataset; (c) the imbalanced training set and (d) the imbalanced test set.
(PDF)

**S2 Table. Tables of evaluation results for all and individual points for the avian specimen dataset.** After accounting for occluded views of some body regions, the sample sizes across labelled points were: N(reflectance standards 1–5) = 5094, N(Throat, Breast, Belly, Flight feathers) = 1698, N(Mantle) = 1697, N(Coverts) = 1696, N(Crown, Nape) = 1695, N(Tail) = 1678 and N(Rump) = 1422.
(PDF)

**S3 Table. Tables of the t-test results comparing pixel distances between predictions using the balanced training set and the imbalanced training set.** These results only include the predictions on bird body regions. We used two-tailed t-test. T-test results with p-values less than 0.05 are in bold text. N is the number of predicted body regions used in the t-tests.
(PDF)

**S4 Table. A table of the t-test results on leave-one-order-out tests for each missing order.** We compared the pixel distances between using the balanced training set (690 images), and the leave-one-order-out balanced training set (660 images) on the predictions (15 images per order) from the missing training orders. We used one-tailed t-test, assuming predictions from the balanced training have less pixel distance than the predictions from the leave-one-order-out balanced training set. T-test results with p-values less than 0.05 are in bold text. N is the number of predicted body regions used in the t-tests.
(PDF)

**S5 Table. ANOVA results on pixel distances using different network architectures and input resolutions for the avian specimen dataset.** The tested networks are Stacked Hourglass and CPM. The tested input resolutions are 494 x 328, 329 x 218 and 247 x 164 pixels.
(PDF)

**S6 Table. ANOVA results on pixel distances of overall and individual points across the original and four tested low-quality avian specimen datasets.** The four low-quality datasets are: (i) rotation (angles between -45˚ to 45˚), (ii) translation on both x and y axes (-500 to 500 pixels), (iii) horizontal flip 50% images randomly, (iv) the combination of all three transformations.
(PDF)

**S7 Table. MANOVA results on the effect of ecotypes and labelling methods (i.e. experts or the Stacked Hourglass) on shape variation (PC1-8).**
(PDF)

**S8 Table. Group distances among the combinations of ecotypes and labelling methods.** (PDF)

## Acknowledgments

We thank M. Adams, H. van Grouw, R. Prys-Jones, and A. Bond from the Bird Group at the NHM, Tring for providing access to and expertise in the bird collection. We thank Z. Varley, L. Nouri, C. Moody and M. Jardine for collecting the avian specimen images. We thank R. Butlin, Z. Zagrodzka and J. Larsson at the University of Sheffield for collecting and providing expertise and access to the Littorina shell images.

## Author Contributions

**Conceptualization:** Yichen He, Steve Maddock, Gavin H. Thomas.

**Data curation:** Yichen He, Christopher R. Cooney, Gavin H. Thomas.

**Formal analysis:** Yichen He, Christopher R. Cooney.

**Funding acquisition:** Christopher R. Cooney, Gavin H. Thomas.

**Investigation:** Yichen He, Christopher R. Cooney.

**Methodology:** Yichen He, Christopher R. Cooney, Steve Maddock, Gavin H. Thomas.

**Project administration:** Gavin H. Thomas.

**Resources:** Gavin H. Thomas.

**Software:** Yichen He, Christopher R. Cooney.

**Supervision:** Christopher R. Cooney, Steve Maddock, Gavin H. Thomas.

**Validation:** Yichen He.

**Visualization:** Yichen He, Christopher R. Cooney.

**Writing – original draft:** Yichen He.

**Writing – review & editing:** Christopher R. Cooney, Steve Maddock, Gavin H. Thomas.

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
