## [Decision Letter · Decision Letter 0]

20 Sep 2022

Dear Dr He,

Thank you very much for submitting your manuscript "Using pose estimation to identify regions and points on natural history specimens" for consideration at PLOS Computational Biology.

As with all papers reviewed by the journal, your manuscript was reviewed by members of the editorial board and by several independent reviewers. In light of the reviews (below this email), we would like to invite the resubmission of a significantly-revised version that takes into account the reviewers' comments.

I agree with both reviewers that this is a very interesting study and method that potentially has high value. However, I share reviewer 2's concern about the representativeness of the training data. I could envision three possible responses:

1.) Perform a series of cross-validation experiments where the data set is split such that the training data has equal representation across the taxa. You could also perform leave-one-out cross validation where each taxon is left out completely from the training data set. These results would pretty clearly demonstrate whether the data bias matters.

2.) Secure additional data to more evenly balance the taxonomic representation in the training data. Obviously this would require a substantial amount of work and may functionally be impossible. I only suggest this on the off chance it's more straightforward for the authors than it seems.

3.) Carefully re-frame the manuscript to focus on the methods, as opposed to any interpretation from a biological perspective and/or clearly state that the generalizability of the method is left for future work.

My advice would be to take a hybrid approach between suggestions 1 and 3, but I will of course leave that to the discretion of the authors.

We cannot make any decision about publication until we have seen the revised manuscript and your response to the reviewers' comments. Your revised manuscript is also likely to be sent to reviewers for further evaluation.

Sincerely,

Samuel V. Scarpino

Academic Editor

PLOS Computational Biology

James O'Dwyer

Section Editor

PLOS Computational Biology

I agree with bot reviewers that this is a very interesting study and method that potentially has high value. However, I share reviewer 2's concern about the representativeness of the training data. I could envision three possible responses:

1.) Perform a series of cross-validation experiments where the data set is split such that the training data has equal representation across the taxa. You could also perform leave-one-out cross validation where each taxon is left out completely from the training data set. These results would pretty clearly demonstrate whether the data bias matters.

2.) Secure additional data to more evenly balance the taxonomic representation in the training data. Obviously this would require a substantial amount of work and may functionally be impossible. I only suggest this on the off chance it's more straightforward for the authors than it seems.

3.) Carefully re-frame the manuscript to focus on the methods, as opposed to any interpretation from a biological perspective and/or clearly state that the generalizability of the method is left for future work.

My advice would be to take a hybrid approach between suggestions 1 and 3, but I will of course leave that to the discretion of the authors.

Reviewer's Responses to Questions

**Comments to the Authors:**

Reviewer #1: Comments for the authors:

First, I would like to thank the author for their work.

Please if any of my comment sounds out of subject because it is due to my misunderstanding, I apologize in advance. Thus it also mean that this specific parts where I misunderstood maybe require some clarification.

This interesting study aims at using deep learning models and more specifically pose estimation to identify body regions and colour of these regions in birds as well as points on snail shells. The manuscript consider multiple network architecture as well as input to identify the best parameters to train such models. The details in the “Model evaluation” section were appreciated. Each evaluation criteria was justified.

Minor issues:

Line 167. “The sample of 1698 bird species includes representatives of 81% of bird genera and 27 bird orders, so the labelled images should capture much of the extent of variation in plumage colour, patterns, and body shape among birds.”. However, one important information for deep leanring training is if there is some class imbalance in the training dataset. S2 Table show as the “N” per class, that there is a high imbalance between classes (come classes represneted with 3 pictures and other up to 3405 pictures). And this can definitly impacted the network performances.

Linked with the previous comment, the objective to test “the relationship between bird taxonomy and network performance to assess whether performance varies among groups of bird species due to broad differences in size, shape and colouration of specimens” (Line 231) is directly impacted by class imbalance. Indeed, if the in the training data set there are birds’ orders that are quite singular and rarely represneted (for example the “Sphenisciformes” N=3), then the model will simply trained and adapt its weights only 3 times per epoch to try to fit to this order. So in order to access whether the performance varies among groups of bird species, I would suggest to either increase the dataset to have approximately the same number of pictures per birds’ orders (but I understand that this may be a lot of work). Alternatively, I would suggest to implement some class weights in the model or to use data augmentation that would still keep the standardize point of view (cropping, zooming, slightly blurring the picture, ...).

Line 228 “We used only the training and validation sets so that every image from the labelled dataset can have a predicted point from the same data partition routine.” Here the justification of why not using a test set was not very clear. However, I have some concern about using only training and validation sets with the cross validation. For wht I understand in the text, the 5094 pictures were used for the cross-validation (all the run having 80% training and 20% validation). If it is not the case and that the 20% of validation set were never seen by the model during the training phase then it’s alright. However, if all the pictures were used during the training, then when evaluation the model what is evluated is the capacity of the neural network to learnt “by heart” the features on this pictures, but not how to recognize the features on pictures. As an example, if you train a student with an exercise where the solution is given at the end, if when evaluating the student you give the exact same exercice, then only the capacity to remember the exercice will be evaluating, not his capacity to understand how to fix the exercice. As a researcher, I would be interested to know if this model as it is right now, would work on a species that was not used during training. Thus I would nevertheless suggest to have a small test dataset (200-300 pictures to access the capacity of the model to detect regions and colours of totally new pictures (either same species and new individuals or even new species).

This may also explain why “the model does not predict labels from non-trainers as reliably” (Line 474). Indeed, if the models was trained using one dataset, it has learn by heart how to deal with these pictures and since the training was done using the trainer annotation, it would be indeed quite close from the trainers’ annotation. To test the differences between predictions, trainer annotations and non-trainer annotation, I would suggest to use the test dataset mentionned previously as a completely independant dataset.

All of this do not make incorrect the main conclusion of deep leanring being able to detect regions on pictures. The great novelties that this study brings, are it’s application for biological questions and the use of large museums datasets. The previous comments are just here to balance some of the results and discussion.

Line 514-521. The results are presented using PC axis, but I would have appreciated to maybe have a biological results as well. Such as wich one of the landmarks would account for most of the variability in snail shell shape between the differents ecosystems. However, this do not really make the paper not understandable.

Line 569. One of the discussion of the paper it to encourage researchers to “manually checking the predictions”. I have the feeling that this is in contradiction with the main argument for using deep learning as set in the introduction. The objective was to create a tool that will help gain time. I do not say that the predictions are not to be checked, but checking manually thousands of predictions (if the dataset is large) can be quite time consuming. The all point of evaluating the model is to evaluate if the model can be trusted regardless a certian amount of errors. If the overall accuracy of the model is not enough then, the model sould not be trusted and would need more fine tuning. However if the model is accurate enough, then maybe using a threshold on the confidence score would be a first step to select the potential wrong predictions that would need to be manually checked (but it would be a small subset of the whole dataset).

Comments:

Line 163. Are the non-labelled regions still detected somehow by the neural network ? I wonder if the network, after trained to detect different body regions on animal with variation in plumage colour, patterns and body shape, can outperformed humans and detect regions that are difficult for humans to annotate (This is purely a personal interest and I am not expecting for this to be updated in the manuscript).

Line 763. S2 Fig. No (a) and (b) on the figure but this is just a small thing to adjust.

Reviewer #2: The review is uploaded as an attachment.

**Have the authors made all data and (if applicable) computational code underlying the findings in their manuscript fully available?**

Reviewer #1: Yes

Reviewer #2: Yes

PLOS authors have the option to publish the peer review history of their article (what does this mean?). If published, this will include your full peer review and any attached files.

Reviewer #1: No

Reviewer #2: No
---

## [Decision Letter · Decision Letter 1]

20 Dec 2022

Dear Dr He,

Thank you very much for submitting your manuscript "Using pose estimation to identify regions and points on natural history specimens" for consideration at PLOS Computational Biology. As with all papers reviewed by the journal, your manuscript was reviewed by members of the editorial board and by several independent reviewers. The reviewers appreciated the attention to an important topic. Based on the reviews, we are likely to accept this manuscript for publication, providing that you modify the manuscript according to the review recommendations.

Sincerely,

Samuel V. Scarpino

Academic Editor

PLOS Computational Biology

James O'Dwyer

Section Editor

PLOS Computational Biology

Reviewer's Responses to Questions

**Comments to the Authors:**

Reviewer #1: I would like to thank the authors for this revised version of the manuscript and the answers to the comments of the editor and the reviewers. 

The manuscript presents lot of improvements: 

- The method section “Specimen imaging and annotation” is much clearer than before about the composition of the different dataset (the balanced, imbalanced, 5094 genus-level). It is much more understandable about what was done and used to create the models. 

- Comparing the effect of balanced vs imbalanced training set on similar balanced and imbalanced testing set is good in my opinion. This will really help to emphasize the importance of taking time to think and create the appropriate training et before running deep learning models. 

- The model comparison of the leave-one-order-out is a nice improvement. 

However, I still have some comments and questions about few specific parts of the study. 

Line 262. Regarding my previous comment: “For what I understand in the text, the 5094 pictures were used for the cross-validation (all the run having 80% training and 20% validation). If it is not the case and that the 20% of validation set were never seen by the model during the training phase, then it’s alright.”. You answered that “the validation images were not seen by the model during training”, but that’s not how the cross validation works and not how you wrote it in the manuscript. You divided the 5094 pictures dataset in 5 batch and each round (there are 5 rounds as you used five-fold cross-validation) you choose one batch as the validation and the rest as the training, and you change every time. Meaning that at the end, all the 5094 pictures are seen by the model to train and to evaluate (line 268 “every image from the 5094 genus-level set can be predicted and evaluated once”). So, all the validation images were seen by the model during training, not at the first cross validation but at the second one or until the fifth round. 

I understand that you want to do that way because you only have one specimen per genus and thus you want the model to see all the genus in the dataset. 

In the manuscript it is well written but your answer to my comment was different. Therefore, I explained the difference here. 

One comment of the previous revision was to create a new dataset with images that the model has never seen. I think that is what you did with the new analysis of balanced vs imbalanced training set, were you have the training and the test that are pictures never used to validate the model at the end of each epoch. 

So, for me it is ok to not have a test set for the model using the 5094, but I would suggest to detailed that you do not need it, as you have been testing the imbalanced training set with both balanced and imbalanced testing set. 

Line 299. The previous comment leads to “After training, validation images were fed into the trained network to generate prediction heatmaps.” 

I understand this sentence and the process for the balanced and imbalanced dataset: training is used to train and the test that are completely new pictures are the validation images in this sentence. 

But I did not understand then what are these validation images for the 5094 genus-level model. Are they the 5094 genus-level pictures? but they are the one used for training so performing predictions on the same pictures used to train is not ideal. Or are they other pictures? 

Line 720 (and 578). “So using a training set that is representative of the full data could again be an optimal solution for datasets that are either taxonomically balanced or imbalanced.”. I quite disagree with this conclusion. I think that showing this comparison and these results is important. But I do not see how to conclude that it is important to have a training set representative of the full data either balanced of imbalanced. In your answer to the reviewers’ comments, you wrote: “These analyses showed that the balanced training set performed well on both the balanced and imbalanced test sets, the imbalanced training set only performed well on the imbalanced test set.”. Then wouldn’t it be more useful to always use a balanced dataset? 

If later one research team, which is focusing on Eurypygiformes for example, wants to use your model to detect the colour of body regions of hundreds of specimens, they may be able to use the model trained using the balanced set, but it would not work if they use the model trained with the imbalanced set. 

Therefore, regarding your results, I would rather conclude to always use a balanced dataset for training, to reach good prediction levels on both balanced and imbalanced new dataset to predict.

Reviewer #2: I appreciate the effort of the authors in clarifying many of the points that were raised and I think that the manuscript improved significantly. I have no major comments and I'm confident that this manuscript will make a significant contribution to make data obtained from biological specimens more accessible.

Even though I think that the manuscript is almost ready for publication, I would only like that the authors clarify the balanced vs. imbalanced dataset analyses and conclusions.

Training with an imbalanced dataset can lead to low accuracy for under-represented classes because of the lack of examples to learn from. This is well illustrated in the leave-one-order-out analyses in which 2 out of 23 orders were under-performed when not present in the training dataset (table 4S). Even though the authors seem to depreciate this finding (“only 2 out of 23 orders were significantly worse when not having images from these orders in the training set.” in their reply; and it is not well explain the practical implications of this in the discussion). This finding shows exactly the limitation of not having a good representation of all orders in the dataset: that the average (across all orders in an imbalanced test dataset) performance of the model might not necessarily be extrapolated to all types of data that are not present (or are under-represented) in the training and test dataset. In lines 690-697 the authors address this result, however, they only present potential solutions (e.g. data augmentation), but ultimately they should also point out that it is not a good idea to blindly predict points for a specific order that was under-represented (or absent) in the training and/or test dataset.

Another important point that is not clear regarding imbalanced datasets is the misinterpretation of the evaluation of the model’s performance if the test dataset is imbalanced. For example, in the results it is stated that “the pixel distance between expert and predicted points averaged 47.3 (1.4% of the image height – 3280 pixels) across all 15 points (10 body regions and 5 reflectance standards)” (line 456-457). However, if the test dataset is not balanced and instead it is representative of the imbalanced full dataset this value of 1.4% would be mostly driven by the performance of the model in predicting points for Passeriformes (as this represents 60% of the data). This means that one cannot use your trained model to predict points for, for example, Leptosomiformes and expect an error of 1.4%. This has to be clearly explained in the discussion.

Finally, while I think that the analyses comparing a balanced and imbalanced training datasets and evaluate the performance of the resulting models on a balanced test dataset is a very useful illustration of the problem of imbalanced datasets, I do not agree that showing the inverse (i.e. evaluate on an imbalanced dataset) is useful at all and can actually mislead the readers. The ultimate goal of training a model to perform a task is to perform this task well across all the possible types of data and not to obtain a “good value” of a given evaluation metric (that might result from an imbalanced test dataset and be manly driven by a handful of data types, in this case orders). Having a model that performs well across all types of data is what can avoid a systematic bias in our predicted data and make your model generalisable (e.g. imagine that someone wants to use your model just for Leptosomiformes, the performance of the model for Passeriformes will be irrelevant). Evaluating a model trained on an imbalanced dataset and tested on an imbalanced dataset does not show that is better to use this strategy, it only shows that the model will perform better with more training data (e.g. if Passeriformes are 60% of the data in both training and test datasets, the model has many data to learn from for this order and the evaluation metric will be manly driven by this order alone, and obviously result in a “good model”). While creating a balanced training dataset might in some cases be difficult without excluding some data for the most common orders (which is not the ideal thing to do and unlikely to be a good advice for most situations) and therefore be better to train with an imbalanced training dataset, it is hard to justify testing the model on an imbalanced dataset if the final aim is to generalize the model across all orders. I would therefore remove any analyses that use the imbalanced dataset for evaluation of the models’ performances as this can be misleading to the readers.

Other minor comments:

Line 107 I think that “automated point annotation” sound better as annotation often refers to annotating the data to create a dataset to train a deep learning model.

Line 186 each specimen was only photographed once right? You shouldn’t have the same specimen in the training and test dataset as this will contribute to a higher overall accuracy, but not necessarily represent the generalization capability of the model to predict points in unseen specimens (i.e. it is easier to predict points on a new image from a specimen present in the training dataset than on a image from a completely new specimen). If you used the same specimens on the training and test dataset you should clearly state that and explain why you did not keep them separate.

Shouldn’t all tables that show results per order also include a column in which the pixel distances were corrected for the size of the pictures (e.g. % of the image height) so that we could also compare the error between the different orders?

**Have the authors made all data and (if applicable) computational code underlying the findings in their manuscript fully available?**

Reviewer #1: Yes

Reviewer #2: None

PLOS authors have the option to publish the peer review history of their article (what does this mean?). If published, this will include your full peer review and any attached files.

Reviewer #1: No

Reviewer #2: No

Figure Files:

Data Requirements:

Reproducibility:

References:

---

## [Editor Report · Decision Letter 2]

7 Feb 2023

Dear Dr He,

We are pleased to inform you that your manuscript 'Using pose estimation to identify regions and points on natural history specimens' has been provisionally accepted for publication in PLOS Computational Biology.

Best regards,

Samuel V. Scarpino

Academic Editor

PLOS Computational Biology

James O'Dwyer

Section Editor

PLOS Computational Biology

---

## [Editor Report · Acceptance letter]

17 Feb 2023

PCOMPBIOL-D-22-01076R2 

Using pose estimation to identify regions and points on natural history specimens

Dear Dr He,

I am pleased to inform you that your manuscript has been formally accepted for publication in PLOS Computational Biology. Your manuscript is now with our production department and you will be notified of the publication date in due course.

With kind regards,

Zsofia Freund
